# The Role of HDACs in the Response of Cancer Cells to Cellular Stress and the Potential for Therapeutic Intervention

**DOI:** 10.3390/ijms23158141

**Published:** 2022-07-24

**Authors:** Rahma K. Alseksek, Wafaa S. Ramadan, Ekram Saleh, Raafat El-Awady

**Affiliations:** 1College of Pharmacy, University of Sharjah, Sharjah 27272, United Arab Emirates; u20105711@sharjah.ac.ae; 2Sharjah Institute for Medical Research, University of Sharjah, Sharjah 27272, United Arab Emirates; 3Clinical Biochemistry and Molecular Biology Unit, Cancer Biology Department, National Cancer Institute, Cairo University, Cairo 12613, Egypt; ekram.saleh@cu.edu.eg

**Keywords:** HDACs, cellular stress, immune modulation, HDAC inhibitors

## Abstract

Throughout the process of carcinogenesis, cancer cells develop intricate networks to adapt to a variety of stressful conditions including DNA damage, nutrient deprivation, and hypoxia. These molecular networks encounter genomic instability and mutations coupled with changes in the gene expression programs due to genetic and epigenetic alterations. Histone deacetylases (HDACs) are important modulators of the epigenetic constitution of cancer cells. It has become increasingly known that HDACs have the capacity to regulate various cellular systems through the deacetylation of histone and bounteous nonhistone proteins that are rooted in complex pathways in cancer cells to evade death pathways and immune surveillance. Elucidation of the signaling pathways involved in the adaptive responses to cellular stress and the role of HDACs may lead to the development of novel therapeutic agents. In this article, we overview the dominant stress types including metabolic, oxidative, genotoxic, and proteotoxic stress imposed on cancer cells in the context of HDACs, which guide stress adaptation responses. Next, we expose a closer view on the therapeutic interventions and clinical trials that involve HDACs inhibitors, in addition to highlighting the impact of using HDAC inhibitors in combination with stress-inducing agents for the management of cancer and to overcome the resistance to current cancer therapy.

## 1. Introduction

The deep-rooted understanding of cancer development and the dense complexity of tumorigenesis have been abbreviated into multiple hallmark capabilities of cancer cells. This includes selective support of growth and proliferation, stress response manipulation to drive vascularization, invasion and metastasis, metabolic repairing, an aggressive microenvironment, and immune modulation that facilitates apoptosis evasion. Cancer cells are exposed to multiple types of intracellular and extracellular stresses, which challenge their ability to proliferate and survive. It’s firmly established that stressed cells are normally eliminated by activation of cell death pathways to protect the whole tissue. However, these stress-induced cell death mechanisms are less active in tumor cells. A cellular stress phenotype is one of the dominant hallmarks of carcinogenesis, which can be triggered in cancer cells by different stressors, such as lack of nutrients and oxygen supply, DNA damage, and endoplasmic reticulum stress. These stress conditions are usually induced by diverse changes in the cancer cells and the harsh conditions of the surrounding environments as well as by extrinsic factors such as chemotherapeutic agents, resulting in shifts in the cellular homeostasis. Cellular responses to different stressors could involve re-establishment of cellular homeostasis or adaptation to stress or, in some conditions, induction of autophagy or cell death [1]. This involves an increase in genomic instability and mutation coupled with changes in signaling pathways and gene expression programs, creating a sophisticated network that researchers have been trying to disclose in recent years. The functional mutations in oncogenes and loss of function of tumor suppressor genes during malignant transformation resulted in a disturbance in the cell regulatory function that is associated with exacerbations in cellular stress phenotypes [2]. These phenotypes are derived by genomic and non-genomic changes which could create an intricate network by incorporating pathways/proteins unrelated to the typical cellular stress response pathways to drive the adaptive and protective phenotypes to a variety of environmental stressful conditions such as oxidative, metabolic, proteotoxic, and genotoxic stress. Uncovering the underlying molecular mechanisms by which cancer cells respond to both extracellular and intracellular stress will provide a deeper view to guide the design of more effective targeted therapeutic strategies for cancer. Both tumor location and type can guide the category of stress that the cells are subjected to. Therefore, based on the level and mode of stress, different defense and survival strategies are mounted. To further illustrate, solid tumors occupying restricted space are at higher risk to experience insufficient oxygen and nutrient supply in addition to physical compression forces [3]. In an overall scenario, cancer cells harbor a vast number of genetic and epigenetic alterations including point mutations, deletions, rearrangements, translocations, and transcriptional silencing. In recent years, epigenetics is the most rapidly expanding field in cancer biological research. Epigenetics refers to the heritable changes in gene expression patterns that do not involve alterations in the DNA genetic code itself. Epigenetic remodeling is characterized by two contradictory hallmarks: reversibility and stability. This upgrades the importance of cancer epigenetics due to its reversible nature, which makes it a captivating tool for novel therapeutic strategies and medication design. Different regulatory epigenetic mechanisms have been identified, which are DNA methylation, histone modification, and non-coding RNA mediated processes [4].

DNA methylation is the covalent addition or removal of a methyl group to the cytosine nucleotide in CpG islands of the DNA sequence, which exist in most of the gene’s promoter regions. This modification is mainly regulated by a family of specialized enzymes known as DNA Methyltransferases (DNMTs) [4]. On the other hand, several epigenetic mechanisms modulate the compaction of DNA into a higher order assembly called chromatin. Chromatin is comprised of nucleosomes that are composed of DNA wrapped around an octamer containing cores of DNA packaging proteins called histones. Currently, several histone post-translational modifications are identified involving ubiquitination, phosphorylation, methylation, and finally acetylation. Such modifications can have a deep impact on the interaction between DNA and histones, thus affecting the gene transcription pattern, DNA repair or replication, and chromosomal organization [4,5].

Recent studies have demonstrated that the histone code represents a molecular watchtower for the chromatin landscape in the regulation of transcription factors to switch gene expression on and off. Acetylation of lysine residues is one of the most robust patterns of histone regulation. Histone acetylation neutralizes the positive charge of lysine residue in the histone tail to reduce the binding strength between the histone and the DNA and subsequently unwind the DNA–histone complex. This modification is implemented by a family of enzymes called histone acetyltransferase (HATs), which results in the ease of the accessibility of the transcription factors and consequently activates gene transcription. On the other hand, the function of HATs is counteracted by another group of enzymes called histone deacetylase (HDACs), which are implicated in removing the acetyl groups from histones, resulting in increased ionic interaction between the negatively charged DNA and the positively charged histones. This DNA–histone complex arrangement yields a more compact chromatin which represses the transcription process. In addition to gene transcription, the function of HATs and HDACs is not only limited to histone proteins; they can also target a wide range of non-histone proteins that are involved in different biological processes such as cell-cycle progression, differentiation, and apoptosis [6].

The results of genome-wide analysis revealed a global alteration in histone acetylation in cancer cells, which is due to impairment in the balance between the acetylation and deacetylation of histone and nonhistone proteins. These alterations can be raised from the aberrant functions of HATs and HDACs, resulting from mutations or translocation or from the dysregulation in the expression pattern of HATs and HDACs [7]. Therefore, recent research is progressively unraveling the correlation between the abnormal function of HATs and/or HDACs with the incidence of different types of human cancers. Overall, histone acetylation patterns are being investigated as a therapeutic target because of their significant role in gene expression regulation. The entanglement of chromatin modifications in cancer cells is remarkable and is still an ongoing field of profound investigation. This review encapsulates and highlights the current interpretation and importance of the chromatin remodeler HDACs family in the response of cancer cells to cellular stress and in carcinogenesis.

## 2. Histone Deacetylases (HDACs)

### 2.1. HDACs Classification

In the human genome, the HDACs superfamily consists of 18 members organized into four major classes (Table 1). The classification can be based on structure, function, and subcellular localization, as well as their homology with yeast HDACs and co-factors. Classes I, II, and IV are zinc-dependent metalloproteins in which a zinc molecule is required in the active site as a cofactor. Class III HDACs are nicotinamide adenine dinucleotide (NAD+)-dependent enzymes that require NAD+ instead of Zn^2+^ as a cofactor and have a structural homology to yeast sir2 proteins [8,9]. Class III HDACs are resistant to HDAC inhibitors (HDACIs) and their exact function in the cell cycle and carcinogenesis is currently dubious. While some have been shown to function as oncoproteins, others have been described as tumor suppressors [10]. Class I HDACs are highly identical to yRPD3, a yeast transcriptional regulator. This class includes HDACs 1, 2, 3, and 8 and is present most abundantly in the nucleus. Class II is related to the yeast Hda1 deacetylase enzyme, and it includes HDACs 4–7, 9, and 10. The class II HDACs determine the status of non-histone substrate acetylation, therefore the class is further subdivided into Class IIa (HDACs 4, 5, 7, and 9) that constantly travels between the cytoplasm and the nucleus and Class IIb (HDACs 6 and 10) that is localized mainly in the cytoplasm and is characterized by possessing two deacetylase domains [11]. Class III HDACs are comparable to yeast SIR2 and include seven members of the Sirtuins (SIRT) family. This class of HDACs has the ability to target proteins in the nucleus, cytoplasm, and mitochondria for posttranslational modifications such as acetylation or ADP ribosylation [12]. Finally, Class IV HDACs includes only HDAC 11, which is mainly localized in the nucleus and has a structural similarity to both class I and II, especially to HDACs 3 and 8 [13].

### 2.2. Dysregulation of HDACs in Cancer

The regulatory function of HDACs in gene transcription and protein activity make these proteins an essential player in a wide array of critical cellular signaling pathways through modulating the acetylation of histone and nonhistone substrates. As shown in Figure 1, the aberrant function of HDACs was described to either regulate the oncogenic cell signaling pathway (Figure 1A) or repress tumor suppressor gene activity (Figure 1B) [16,19]. It was reported that the aberrant expression of HDACs can affect the function of proteins involved in the cell cycle, proliferation, differentiation, angiogenesis, invasion, metastasis, and apoptosis [16,20,21,22,23,24]. The overexpression of HDACs becomes well-established in different types of cancer. This is evident with HDAC1 overexpression in prostate cancer and HDAC2 overexpression in gastric, colorectal, and endometrial sarcomas, which is correlated with decreased expression of p21 [25]. In addition, HDAC4 overexpression was investigated in esophageal carcinoma and was found to be significantly correlated with a higher rate of cell proliferation and tumor migration and lymph node metastasis, resulting in a higher tumor pathological grade and lower survival rate [26]. Moreover, Halkidou et al. reported that a high level of HDAC4 is associated with hormone-resistant cases of prostate cancer patients [25,26]. In line with this, the knockdown of HDAC4 in several cancer cell lines was found to stimulate p21 expression and consequently inhibit tumor cell proliferation in vitro and tumor growth in vivo [27,28]. In addition, studies revealed the potential role of the abnormal recruitment of HDACs to specific promoters through the interaction with fusion proteins in hematological malignancies [6,29]. Abnormal recruitment and function of HDACs can be raised from dysregulation in the expression pattern of HDACs. [28,30,31]. Collectively, the inhibition of critical growth suppressive genes by the upregulation of HDACs is a dominant underlying mechanism in the promotion of cancer cell development and proliferation that can be counteracted by the inhibition of HDACs.

Despite the broad range of anticancer effects of HDACIs that propose an oncogenic role of HDACs in tumor development, it has been found that the genetic downregulation of HDACs might have tumorigenic effects. The overexpression of HDAC6 in breast cancer patients at mRNA and protein levels was reported to result in a better prognosis than for those with low levels in terms of survival rates [32]. Another study reported that the reduction in the expression of class II HDAC genes, HDACs 5 and 10, in lung cancer patients was associated with a poor prognosis in which HDAC10 was the strongest predictor of a poor prognosis [33]. Altogether, the dysregulated function of HDACs in cancer can contribute to either tumor promotion or suppression.

## 3. Role of HDACs in Cellular Stress Response

The dependency of cancer cells on divergent pathways in response to different environmental stresses has been well established. This is through triggering various molecular mechanisms that promote genomic instability and mutations, reprogram different metabolic systems, and alter gene expression patterns to escape the growth inhibitory signals and the body’s immune system inspection. A better understanding of the underlying molecular pathways involved in the adaption of cancer cells to different stressors might open a new avenue for more efficacious strategies for cancer therapy [3]. Nowadays, intense research has been directed towards the role of the HDACs family in facilitating tumor cell survival and proliferation under stress conditions through modulation of the acetylation patterns of histones and nonhistone proteins. Interestingly, HDACs were massively recognized as potential therapeutic targets as they can contribute to the abnormal epigenetic conditions associated with the cellular stress response in order to preserve cancer development and overcome apoptotic pathways. The type and the magnitude of cellular response to stress depends mainly on the level and the type of insult. Additionally, the interplay between these response pathways determines the fate of the stressed cell [34]. The HDACs family plays a significant role in the adaptive stress responses that involve genotoxic, proteotoxic, oxidative and metabolic stresses. Below is a discussion of the different types of cellular stress and pathways involved in the response of cancer cells to each type with emphasis on the role of the HDACs family of enzymes.

### 3.1. Genotoxic Stress (DNA Replication Stress, DNA Damage Response, and DNA Repair Pathways)

DNA damage is a crucial factor in the development and progression of cancer. Cancer cells undergo genotoxic stress when they encounter endogenous or exogenous DNA-damaging agents that have a direct or indirect impact on the integrity of their DNA. In the presence of DNA damage, cancer cells respond by activating biochemical repair machinery that leads to either enhancing cell survival or inducing cell death. Thus, incompetent DNA repair is a predominant driving force behind cancer establishment, progression, and evolution [35].

Ataxia-telangiectasia mutated (ATM) protein has a leading role in the DNA damage response. ATM stimulates the activation of the BRCA1, CHK2, and p53 genes, leading to cell cycle arrest and DNA repair through the activation of the CDKN1A (p21), GADD45A, and RRM2B genes [36]. Over-activation of ATM promotes the adaptation of cancer cells to genotoxic stress. Conversely, impaired ATM function exhibits chromatin exposure and augments genomic instability, which enhances sensitivity to DNA-damaging modalities (e.g., irradiation, and chemotherapeutics) [37]. It was demonstrated that selective depletion of HDAC1 and HDAC2 was sufficient to reduce ATM activation, thus toning down the subsequent phosphorylation of BRCA1, CHK2, and p53 and increasing the susceptibility to DNA break induction in several tumor types [36]. Interestingly, the silencing of HDAC4 by RNA interference downregulated the level of 53BP1 protein, a well-known tumor suppressor protein that participates in the early steps of the DNA-damage-signaling pathways, which abrogated the DNA-damage-induced G2/M checkpoint arrest and increased the radiosensitivity of HeLa cells (Figure 2A). Thus, HDAC4 was proposed to have a prominent role in cell cycle regulation after ionizing radiation [38]. Furthermore, yeast SIR3 has been shown to be prominently recruited at various sites of DNA damage. The accumulation of this type of deacetylase has been hypothesized to facilitate DNA repair and to protect the unrepaired DNA ends through induction of compact chromatin alignment [39]. In a similar manner, SIRT1 has been identified as a major player in the DNA damage response, acting as a deacetylase of proteins involved in DNA repair at sites of DNA damage [40]. Moreover, SIRT1 functions as an enhancer of DNMT1, DNMT3B, and zeste homologue 2 (EZH2). These proteins are recruited at sites of DNA double strand breaks and induce histone repressive modifications such as hypoacetylation of H4K16, H3K9me2/me3, and H3K27me3. These histone modifications help in the establishment of the compact chromatin around the damaged site by forming a silencing complex with DNMT3b, polycomb, and a repressive complex of four components (SIRT1 and EZH2) that pairs with γH2AX, forming DNA-damage-induced foci (Figure 2A) [41]. In addition to H4K16 deacetylation, SIRT1 was reported to deacetylate a member of the HAT family called hMOF (human MOF), which consequently affected its recruitment at sites of DNA damage and caused downregulation of DNA double strand break repair genes such as BRCA2, RAD50, and FANCA in human colorectal cancer cells (HCT116) [42]. On the other side, SIRT1, along with E2F1 transcription factor, are stimulated among the signaling cascade initiated by DNA single strand break molecular sensor PARP1 to guide the transcription of ADP-ribosylation factor (ARF), which is one of the crucial genes that are modulated in response to continuous DNA breaks (Figure 2B) [43,44].

In addition to the significant role of HDACs in the DNA damage response, they participate in the regulation of replication and S-phase progression. It was suggested that HDACs operate the progression of the replication fork by inducing global changes in chromatin structure that affect the functions of DNA polymerase. Dysregulation of the chromatin structure results in uncontrolled origin firing and replication fork collapse, which promotes DNA damage and genomic instability, leading to cell death [45,46]. Previously, it was demonstrated that treatment of cutaneous T cell lymphoma (CTCL) cell lines (HH and Hut78) with a selective HDAC3 inhibitor caused a 50% reduction in DNA replication fork velocity and remarkable cell growth arrest [47]. These results were consistent with another study conducted by Srividya et al., which showed that genetic deletion of HDAC3 in mouse embryonic fibroblasts (MEFs) triggered apoptosis and yielded very early embryonic lethality. They attributed these effects to inefficient removal of acetyl residues from the histones, which causes a flaw in chromatin assembly accompanied with persistent DNA damage accumulation that usually occurs during DNA replication and impairs the progression to the G2 phase [48]. The critical role of HDACs in the adaptation to genotoxic stress supports the potential approach of targeting them in rapidly proliferating tumor cells while being nondestructive to the surrounding nonmalignant cells.

### 3.2. Proteotoxic Stress (Heat Shock Response and Endoplasmic Reticulum Stress)

Dysregulated protein homeostasis is one of the emerging processes involved in tumor progression. The rate of protein formation is influenced by transcription, translation, and degradation processes; all of them are regulated by the chromatin arrangement state. Dysregulation in proteostasis results in impaired protein synthesis or misfolded proteins. This triggers endoplasmic reticulum (ER) stress, which could result in an overall degeneration in cellular function [49,50,51]. The ER is the main area for monitoring protein products, where only the correctly posttranslational folded proteins can exit the ER to the Golgi apparatus to be delivered to their distinct destination. Interestingly, a group of proteins called chaperones exist in the ER and cytosol to maintain protein homeostasis by programming the folding of newly synthesized proteins and partially folded proteins and prevents the misfolded protein aggregates [52]. In the tumor microenvironment, hypoxia and nutrient deprivation states induce ER stress. Therefore, cancer cells depend on an interconnected network of proteostasis signaling pathways, such as the unfolded protein response (UPR), to sustain protein stability. The UPR pathway modulates the rate of protein synthesis mainly through interacting with proteasomal systems such as the macroautophagic (autophagy-lysosome) system, aggresomal pathway and heat shock chaperone protein system, to correct impaired protein clearance and folding or to induce apoptosis in persistent ER stress [53].

Cytoplasmic HDAC6 was found to represent a master chief in the regulation of the cytoprotective response to proteotoxic stress through association with proteasomal proteins (Figure 3). For instance, HDAC6 forms a complex with p97/VCP and UFD3/PLAP, which are involved in controlling the ubiquitin/proteasome system. P97/VCP is a chaperone that facilitates the degradation of misfolded proteins when the ubiquitin-dependent proteasomal turnover of proteins is overwhelmed and paralyzed (Figure 3A) [54,55]. In addition, HDAC6 induces the expression of dominant chaperons in response to the accumulation of ubiquitinated protein aggregates. Initially, HDAC6 senses the abnormal accumulation of ubiquitinated misfolded proteins via its ubiquitin-binding activity. Consequently, it promotes the dissociation of a repressive HDAC6/heat-shock factor 1 (HSF1)/heat-shock protein 90 (HSP90) complex, where the liberated HSF1 activates HSP gene expression to induce cell survival (Figure 3B) [20,56,57]. Accordingly, HDAC6 inhibition has been shown to increase the acetylation of HSP90 and suppress its function as a molecular chaperon, which increases the number of misfolded proteins in the cell (Figure 3C). When the rate of misfolded proteins exceeds the processing or folding capacity of protein chaperones, it will result in chronic unresolved ER stress and subsequent apoptosis induction in cancer cells [20]. In addition to HSP90, HDACIs increased the acetylation levels of other chaperones such as regulated protein 78 (GRP78), which causes the induction of protein misfolding and proteotoxic stress, leading to the suppression of cellular proliferation and subsequent apoptosis. On the other hand, HDACIs can induce ER stress in cancer cells indirectly through the upregulation of the reversion-inducing cysteine-rich protein with Kazal motifs (RECK) gene, which is a well-known member of the metastasis suppressor genes that was found to modulate tumor cell invasiveness and metastasis. The upregulation of RECK was found to sequester GRP78, which releases ER transmembrane sensor proteins to eventually induce ER stress and activate apoptosis [52,58,59,60,61]. Collectively, these studies present a new HDAC-targeted approach in limiting metastasis and angiogenesis and in increasing the susceptibility of cancer cells to ER stress through induction of the intracellular proteotoxic environment [62].

### 3.3. Oxidative Stress

Reactive oxygen species (ROS) are byproducts of the normal oxygen metabolism, which serve a critical role in several biological functions, signaling pathways and redox homeostasis. However, the strict regulation of ROS by scavenging systems is compulsory because of their possible toxic impact on cellular structures. Indeed, the impairment that ROS can impose on the cell does not solely depend on their intracellular levels, but also on the equilibrium between ROS and the endogenous antioxidant species. When such an equilibrium is disturbed, oxidative stress is provoked, resulting in severe damage to intracellular biological components such as DNA, RNA, and proteins, which is a recognized hallmark of cancer.

Cancer cells usually exist in a hypoxic microenvironment, which further boosts their metabolic activity and oncogenic stimulation that in turn generates a high level of ROS [63]. Strikingly, cancer cells use several mechanisms, such as activation of ROS-scavenging systems, suppression of cell death factors, and generation of lactate instead of employing aerobic respiration, to adapt to the massive ROS accumulation without disturbing the energy demand of cancer cells to support their proliferation and survival [3,64,65]. There are many antioxidant genes that are associated with cellular responses to oxidative stress including superoxide dismutases (SODs), glutathione peroxidases (GPXs), glucocorticoid receptors, heme oxygenase (HMOXs), and hypoxia-inducible factor-1α (HIF-1 α). Many of these genes have been reported to be regulated by epigenetic mechanisms. One of the most powerful and well-known examples of cellular defense machinery against oxidative damage is the KEAP1-NRF2 pathway, which includes the transcription factor nuclear factor erythroid 2-related factor 2 (Nrf2) and its negative cytoplasmic regulator kelch-like ECH-associated protein 1 (Keap1). Under oxidative and electrophilic stress, Keap1 allows Nrf2 phosphorylation and translocation into the nucleus. In the nucleus, Nrf2 activates the expression of a wide range of antioxidative detoxifying enzymes by binding to the antioxidant response element (ARE) in their regulatory regions that rescues the cell from oxidative injury [66]. Surprisingly, HDACs regulate Nrf2 activity and ARE-dependent gene expression through the direct modulation of Nrf2 acetylation. This was evident by the increased acetylation level of Nrf2 by selective inhibitors of Sirtuin 1 (SIRT1), such as EX-527 and nicotinamide, which results in enhancing the binding of Nrf2 to ARE and thereby stimulating Nrf2-mediated gene expression. In the same line, SIRT1 activators (such as resveratrol), have been shown to deacetylate Nrf2 and to suppress Nrf2 signaling (Figure 4A) [67]. In addition, HDACs and their inhibitors were reported to regulate the Nrf2 pathway via the adjustment of histone acetylation at the promoter regions of antioxident genes. Liu et al. reported that HDAC3 is a negative regulator of the Nrf2 pathway through the function of the p65 subunit of NF-κB, which enhances the interaction of HDAC3 with MafK, a known dimerization partner with Nrf2. This interaction causes the recruitment of HDAC3 to ARE that consequently promotes the maintenance of the histone hypoacetylation state and hence represses ARE-dependent gene expression (Figure 4B) [68]. HDAC1 was reported to work as a corepressor of the transcription factor basic leucine zipper transcription factor 1 (Bach1), which has an important role in repressing the oxidative stress response through forming a complex with p53 and HDAC1 and the nuclear corepressor N-CoR, inhibiting cellular senescence of murine embryonic fibroblasts in response to oxidative stress (Figure 4C) [63,69].

Additionally, the functional role of HDACs in regulating oxidative stress response was demonstrated through HIF-1. HIF-1 is one of the dominant modulators of genes responsive to hypoxia, a common event in solid tumors, that causes an excessive production of ROS, leading to oxidative stress. HIF-1 is composed of two subunits: the hypoxia-regulated HIF-1α and the oxygen-insensitive HIF1ß subunits, which form a heterodimer and bind to hypoxia responsive elements (HREs) in oxygen-regulated genes including VEGF. These genes are involved in cellular biological processes such as angiogenesis and augment oxygen delivery to tumor hypoxic regions. In both human and mouse cell lines, it was reported that class I HDACs, in particular HDAC1 and HDAC3, directly interact with the HIF-1α protein and induce its deacetylation, which enhances its stability and transactivation function under hypoxic conditions. These results actively indicate that the stabilization of HIF-1α protein is accelerated through direct interaction with HDAC1 and HDAC3, leading to enhanced tumor angiogenesis [70]. Similarly, a positive crosstalk was established between HDAC1, HIF-1α, and metastasis-associated protein 1 (MTA1) in which HDAC1 participates in the MTA1-induced stabilization of HIF-1α. It was suggested that hypoxia induces the expression of MTA1, which inhibits the acetylation of HIF-1α by recruiting HDAC1, resulting in the stabilization of HIF-1α and inhibiting its degradation. These findings were further confirmed using the potent HDACs inhibitor Trichostatin A (TSA), which reduced the stability of HIF-1α in breast cancer cell lines. This establishes a close connection between MTA1-associated metastasis and HIF-1-induced tumor angiogenesis through the activity of HDAC1 [71]. In addition to HDAC1 and HDAC3, HDAC7 was reported to be co-translocated with HIF-1α to the nucleus under hypoxic conditions, where it subsequently increases the transcriptional activity of HIF-1α through the formation of a complex composed of HIF-1α, HDAC7, and p300 [72]. However, the underlying mechanism of transcriptional regulation of HIF-1α by HDAC7 is still not fully understood.

For NAD-dependent deacetylases, SIRT1 was reported to modulate different angiogenesis-related genes under oxidative stress, such as membrane-anchored matrix metalloproteinase MMP14 (MT1-MMP), Flt1, CXCR4, Pdgf, and EphB2 [73,74], while SIRT3 loss was associated with an increase in the production of ROS, causing the stabilization of HIF1α. Similarly, in human breast cancer, the reduction in the expression of SIRT3 results in upregulation of the HIF-1α target genes. These findings highlight the role of SIRT3 in the hypoxic response of tumor cells, exposing a potential area for therapeutic intervention [75,76]. To sum up, HDACs have ab influential role in many oxidative stress pathways, including both sensing and coordinating the cellular response to oxidative stress pathways, and HDACIs might be certified candidates for targeting oxidative stress pathways.

### 3.4. Metabolic Stress (Hypoglycemia and Hypoxia)

Metabolic stress is a common phenomenon in human tumors. It results from insufficient nutrient supply to tumors, which is caused by angiogenesis deficiency and elevated metabolic demands due to aggressive, uncontrolled cellular proliferation. While normal tissues have restricted cell division that is strictly regulated by growth factors and nutrient availability, tumor cells lack this control of cell division. Nevertheless, they proliferate independently of restricted nutrient supply by relying on the incompetent glycolysis process as an energy supply source, which further exaggerates their metabolic stress status [77]. Under normal conditions, these stressful factors drive the normal cells to metabolic catastrophe, leading to the termination of cell proliferation and growth. On the other hand, cancer cells acquire some genomic and metabolic phenotypes that help them to grow and escape the apoptotic pathways stimulated as a result of modifications in the tumor microenvironment [78]. The most dominant metabolic phenotype of cancer cells, which is an essential step in the adaptation to metabolic stress, is the elevation of glucose uptake and the production of lactate for aerobic glycolysis regardless of oxygen presence [79]. Additionally, alternative carbon and energy sources, such as fatty acids and amino acids, are used by cancer cells to fulfil increased energy demands and to respond to the various metabolic stresses and oncogenic signaling [80,81].

Protein acetylation levels were reported to be affected by cellular metabolism through the regulation of NAD+ and acetyl-CoA concentrations. Thus, HDACs have a pronounced role in the metabolic reprogramming in cancer cells [82]. Indeed, several reports uncovered the role of the SIRT family in manipulating several metabolic pathways (Figure 5). For instance, SIRT3 and SIRT6 suppress tumorigenesis by inhibiting aerobic glycolysis or a glycolytic switch (Warburg effect) through the destabilization of HIF-1α and inhibition of glycolytic kinases [76]. SIRT6 was also found to inhibit gluconeogenesis, which generates glucose from noncarbohydrate precursors, through the deacetylation of the transcription factor FoxO1, leading to its export to the cytoplasm. The nuclear exclusion of FoxO1 reduces the expression of phosphoenolpyruvate carboxykinase (PCK1) and glucose-6-phosphatase (G6PC), which are rate-limiting enzymes in gluconeogenesis (Figure 5A) [83]. Furthermore, SIRT6 was shown to induce the deacetylation of pyruvate kinase M2 (PKM2), a glycolytic enzyme, which results in its nuclear export through exportin 4 [84]. Interestingly, PKM2 was found to have HDAC3-dependent regulation of the expression of oncogenes such as c-Myc and cyclin D, which promotes tumorigenesis [85].

Despite the negative regulation of glycolysis by SIRT6, SIRT3 and SIRT5 were found to contribute to cancer cell proliferation and survival in diffuse large B cell lymphoma and breast cancer by regulating the function of metabolic enzymes [86,87]. The depletion of SIRT3 in large B cell lymphoma blocks glutamine flux to the tricarboxylic acid (TCA) cycle through inhibition of glutamate dehydrogenase that results in the reduction of acetyl-CoA pools, which causes the induction of autophagy in cancer cells (Figure 5B) [87]. Similarly, the overexpression of SIRT5 in breast cancer protects the mitochondrial enzyme glutaminase (GLS) from ubiquitin-mediated degradation through SIRT5 dependent-desuccinylation of lysine164 residue, which stabilizes GLS [86]. Indeed, SIRT5-dependent GLS stabilization is the main mechanism by which SIRT5 promotes cancer cell growth and survival. The involvement of SIRT2 in supporting tumorigenesis through modulating metabolic pathways was also reported. SIRT2 regulates cellular metabolism and metastasis in colorectal cancer through deacetylation of isocitrate dehydrogenase 1 (IDH1), which plays an important role in glutamine metabolism. The increased deacetylation level of IDH1 at lysine 224 simulates its enzymatic activity and subsequently induces the generation of NADPH and glutathione (GSH), which protects cancer cells from ROS produced during their rapid proliferation rate (Figure 5C). In addition, IDH1 stimulation by deacetylation induces the proteasomal degradation of HIF-1α, which exerts a suppressive effect in colorectal cancer metastasis [88].

HDACs were found to be involved in regulating the covalent attachment of fatty acids to proteins, which is known as fatty acylation of proteins. This protein modification is known to be essential in membrane synthesis and cellular signaling during cancer growth and progression. Surprisingly, some HDACs established a greater catalytic activity towards acyl groups when compared with acetyl peptides [89,90]. For instance, the catalytic efficiency of HDAC11 as a lysine defatty-acylase was reported to be more than 10,000-fold higher than its deacetylase activity [89]. Recently, SHMT2α, a mitochondrial enzyme involved in one carbon metabolism and found to exert a critical metabolic function in cancer cells, has been identified as a substrate of the lysine defatty-acylase activity of HDAC11. It was demonstrated that HDAC11 removes the acyl groups from the SHMT2α surface, which prevents its translocation to late-lysosome/endosome. This effect leads to the polyubiquitylation and degradation of type I interferon receptor chain 1 (IFNαR1), which downregulates the IFN signaling that is involved in metabolic reprogramming [7]. Consequently, elevenostat, which is an HDAC 11 inhibitor, represents a potential treatment approach that targets metabolic lipid dysfunction in cancer [91]. Overall, HDACs exert a critical role in regulating glucose homeostasis and energy balance and strengthening the metabolic phenotype of cancer cells to promote their survival regardless of the intracellular nutrient stress environment.

## 4. HDACs and Anti-Tumor Immune Response

Different cellular stress conditions were recognized to induce autophagy (ATP secretion) or necrosis (inflammation) or to stimulate the release of damage-associated molecular patterns (DAMPs), which trigger a form of cell death called immunogenic cell death (ICD), to eliminate the stressed cells. ICD triggers the activation of the immune response against cancer cells through the attraction of professional antigen-presenting dendritic cells and the subsequent priming of cytotoxic T lymphocytes. However, it is well known that cancer cells are resistant to antitumor immunity due to the highly immunosuppressive tumor microenvironment that favors the immune escape of cancer cells [92,93]. Indeed, ICD can be induced by certain conventional chemotherapeutic drugs through increasing the immunogenicity of cancer cells to reactivate anticancer immunity. Accumulating evidence suggests that HDACs can regulate the tumor microenvironment and modulate the anti-tumor immune responses, which affects tumor progression. Interestingly, HDACIs can enhance the antitumor immunity by facilitating the production of proinflammatory cytokines such as interleukin 6 (IL-6), IL-8, IL-1β, macrophage inflammatory protein 1 (MIP1), tumor necrosis factor-α (TNFα) and IFNγ, which activate more immune cells such as T lymphocytes [94,95]. On the other hand, the selective inhibition of HDAC6 demonstrated a crucial role in inducing the activation of T cell functionality and immunosurveillance through decreasing the production of anti-inflammatory cytokine IL-10 [96,97,98]. Inhibition of HDAC8 was shown to block Hepatocellular carcinoma tumorigenicity in a T-cell-dependent manner and this effect was abrogated by CD8^+^ T cell depletion or regulatory T cell adoptive transfer [99]. A low dose of the HDACi trichostatin-A was shown to enhance the anti-tumor effects of immunotherapies by modulating the suppressive activity of infiltrating macrophages and by inhibiting the recruitment of myeloid-derived suppressor cells in different types of tumors [100]

In addition to cytokine production, HDACIs induce changes in the expression of major histocompatibility complex (MHC) class I, which presents a diverse number of peptides to cytotoxic T lymphocytes to induce its activation [96,101]. Indeed, the histone deacetylation at the promoter sites of genes which produce the components of antigen processing machinery (AMP) was found to be dysregulated in multiple types of cancer such as colon cancer, esophageal squamous carcinoma, renal cell carcinoma, and melanoma. This dysregulation of histone deacetylation could be reversed by treatment with HDACI [102,103]. For instance, treating metastatic carcinoma cells with TSA increased the expression of APM components such as transporter associated with antigen processing 1 (TAP-1), TAP-2, low-molecular-weight protein-2 (LMP-2), and Tapasin, as well as the cell surface expression of MHC class I, which boost the susceptibility of cancer cells to cytotoxic T lymphocytes-mediated target cell killing [101,104]. Moreover, immune cells such as natural killer and cytotoxic T cells become more activated after inhibition of class I HDACs by entinostat [105]. In comparison to the above-mentioned role of HDACIs in inducing the immune response, several HDACIs such as SAHA, LBH589, and TSA were found to suppress both adhesion and costimulatory molecules such as CD40, CD80, and CD83 on dendritic cells, which attenuate T lymphocyte proliferation and cytokine secretion [106,107,108]. Furthermore, it was reported that HDACs are involved in the transcriptional regulation of programmed death ligand-1 (PD-L1), which is a transmembrane protein that engages with PD-1 on immune cells to inhibit the antitumor immune response. HDAC5 was reported to regulate the expression of PD-L1 through direct interaction with NF-κB p65 and inhibition of HDAC5-sensitized tumor cells to immune checkpoint blockade [109]. The upregulation of histone acetylation at the PD-L1 gene was induced with HDACI treatment in melanomas cells, which increased the expression of PD-L1 and, in turn, blocked immune surveillance [110]. Collectively, HDACIs established the ability of HDACs to modulate the immunogenicity and antigen-presenting capacity of tumor cells, which affect the anti-tumor response of the immune system.

### Immunomodulatory Role of HDACs under Genotoxic Stress

The suppression of the immune system in cancer patients is one of the major factors that facilitates tumor progression and resistance to chemotherapy and radiotherapy. Therefore, the approach of boosting the immune activity by a combination of chemotherapy and radiotherapy with additional therapeutic drugs to enhance their antitumor effect has become a promising strategy in cancer therapy. Indeed, a continuous activation of variable immunoreceptors was recognized in immune cells in the surrounding tumor environment. However, the inadequate expression of immunoreceptor ligands and the immunosuppressive nature of the tumor microenvironment limit the activation of anti-tumor immunity. Among the several immunoreceptors that are involved in the immune response shield in cancer is natural-killer group 2, member D (NKG2D). NKG2D is expressed in different types of immune cells such as natural killer and T lymphocytes cells, and it is cytotoxic to cancer cells [96,98]. The expression of the members of NKG2D ligands, such as major histocompatibility complex class I-related chain A and B (MICA/B), was reported to be increased by DNA damage induced by ionizing radiation (IR) or chemotherapeutic agents. This increase in the expression of NKG2D ligands potentiates the killing of cancer cells by NK cells [111,112,113]. However, in some cancer cell lines, the expression of MICA/B wasn’t induced after DNA damage, which was suggested to be due to chromatin disorganization that affects their sensitivity to DNA-damaging agents. Interestingly, it was demonstrated that the inhibition of HDACs activity restored and enhanced DNA-damage-induced MICA/B expression in the resistant cells. This observation suggests that HDACs may contribute to the enhancement of immune activation following DNA damage induced by radiotherapy or chemotherapy [114,115].

## 5. Therapeutic Implications of HDAC Inhibitors

Various studies over the past decade have displayed HDACs as a crucial player in cancer development and progression by reversibly regulating the acetylation level of both histone and nonhistone proteins. This is unlike the permanent and irreversible cancer-associated genetic aberrations such as the overexpression of oncogenes and suppression of tumor suppressor genes. Abnormal HDACs expression and recruitment has been reported in different human cancers, highlighting them as a significant target against cancer. Mechanistically, HDACs suppression was found to be correlated with the regression of tumor growth through modulating different mechanisms including inhibition of angiogenesis and activation of apoptosis [116,117]. The currently available HDACIs are classified into four classes: hydroxamates (e.g., suberoylanilide hydroxamic acid (SAHA)), benzamides (e.g., MS-275), cyclic peptides (e.g., romidepsin), and aliphatic acids (e.g., valproic acid). In addition, HDACIs can be further classified based on their specificity to HDAC isoforms.

A great investigation has been previously directed toward the development of the first-generation non-specific HDACIs, which were mainly pan inhibitors that targeted multiple HDAC isoforms. The hydroxamate class agent called vorinostat (SAHA) was the first nonspecific HDACI that was approved by the FDA for the treatment of cutaneous T cell lymphoma (CTCL). In preclinical studies, SAHA demonstrated several anticancer mechanisms such as the induction of apoptosis and cell cycle arrest in cancer cells [118]. Another two nonspecific hydroxamate class agent HDACIs called belinostat and panobinostat received FDA approval in 2014 for the treatment of peripheral T-cell lymphomas (PTCL) and multiple myeloma, respectively. However, these pan HDACIs endure serious limitations such as secondary effects including cardiac toxicity, gastrointestinal side effects (anorexia, diarrhea, nausea, and vomiting) and hematological effects (anemia, lymphopenia, and thrombocytopenia), in addition to the lack of efficacy against solid tumors when used as single agents compared to hematological malignancies [119]. Accordingly, recent advancements are focused on overcoming these hindrances by enhancing HDACI isoform selectivity through targeting cap groups around the catalytic site [120,121,122,123,124].

Currently, next-generation HDACIs are developed as isoform-selective HDACIs and are tested in preclinical settings, such as tubacin (a selective HDAC6 inhibitor) and PC-34051 (a selective HDAC8 inhibitor) [125]. A recent study demonstrated the ability of a novel selective HDAC6 inhibitor, azaindolylsulfonamide (MPT0B291), to reduce cell viability and to increase the cell death of human and rat glioma cell lines but not normal astrocytes. Moreover, MPT0B291 is reported to induce cell death and cell cycle arrest, which suppresses cell proliferation in the C6 and U-87MG cell lines (in vitro) and in xenograft as well as allograft animal models (in vivo). Mechanistically, MPT0B291 was found to increase p53 acetylation and its subsequent activation. In turn, activation of p53 induces senescence and apoptosis by controlling its target genes, p21, Bax, and PUMA, which leads to cell cycle arrest and the inhibition of proliferation [126]. On the other hand, preclinical studies suggested a potent efficacy of an HDAC8-selective inhibitor, PCI-34051, against T cell lymphomas [127]. The mechanism of action of PCI-34051 was demonstrated to involve phospholipase C gamma 1 (PLCγ1) activation, which is a signal transducer following T cell receptor activation, and calcium-induced apoptosis in T cell lymphomas [128]. Interestingly, the use of an HDAC8-selective inhibitor may have a better effect in solid tumors since its knockdown inhibits lung, colon, and cervical cancer proliferation [129].

In spite of the great achievements and rapid development of the isoform-selective HDACIs, further investigations are needed to compare these selective inhibitors with the well-known pan inhibitors in terms of drug efficiency and side effects. In addition, it is important to identify the cancer-associated HDACs and to consider them in the strategy of designing selective HDACIs. This knowledge not only will lead to the development of more selective and less toxic HDACIs, but it will also illustrate the essential biological pathways for HDACIs to exert their anti-cancer activity, thereby improving the rational design of new anti-cancer therapies. Moreover, a novel perception in the HDACIs formulation field involved combining the features of inhibiting protein kinases and HDACs in a single molecule. These compounds may allow not only blocking the catalytic sites of HDACs and kinases, but also preventing their binding with other proteins independently of their catalytic activities [130]. Recently, mouse models carrying catalytically inactive HDACs, which mimic a specific HDACI treatment, were generated to be used as a valuable tool to investigate the function of particular HDACs in certain cancer types [28].

### 5.1. Clinical Trials of HDACs Inhibitors

Over the past decades, enormous studies have been conducted to characterize the effects of HDACIs in the underlying tumor biological mechanisms such as transcriptional regulation, metabolism, angiogenesis, DNA damage response, the cell cycle, apoptosis, and protein degradation. The results of these studies indicate that HDACIs have potential anti-tumor activity with versatile anticancer effects. Therefore, intensive clinical trials using multiple HDACIs have been conducted for treatment of both hematological and solid malignancies as a single anticancer drug or in combination with other anti-cancer therapeutics [131]. Among them are the clinical trials of FDA-approved inhibitors such as vorinostat, romidepsin, and belinostat, which were performed to check for their efficiency in different types of hematological and solid malignancies. Despite the promising responses and safety profile that were reported from two phase II studies in refractory indolent follicular lymphoma, a modest effect was obtained in solid tumors such as breast [132], colorectal, ovarian and peritoneal [133,134,135], and prostate cancers [136,137]. Other non-FDA-approved HDACIs such as CI-994 underwent clinical trials as single agents in phase I/II and in combination with gemcitabine and capecitabine in phase I for treatment of solid tumors [138,139,140]. An orally active HDACI (ITF2357) was found to reduce inflammatory cytokines production. It was investigated in phase II trials on pretreated refractory Hodgkin’s disease patients [141]. Multiple models have been observed and summarized in Table 2. Although the utilization of HDACIs opened the way for a new class of anti-cancer drugs, they are subjected to some limitations such as resistance and toxic effects. Future investigations using multicenter clinical trials are recommended to provoke approaches for enhancing their selectivity to augment their accumulation in cancer cells even with lower doses, in order to decrease the unwanted side effects and off-target effects.

### 5.2. HDAC Inhibitors in Combination Therapies

Combination cancer therapy, which is a modality that combines two or more therapeutic agents, is considered the cornerstone in cancer therapy. The preference for the combination therapy approach in combating cancer is due to its ability to limit drug resistance, the cancer stem cell population, tumor growth, and metastasis, while in parallel providing a synergistic or additive manner in targeting cancer pathways., in addition to the use of a lower therapeutic dosage of each individual drug in the combination regimen, which, in turn, lowers the off-target side effects [151]. Many chemotherapeutic agents exert their cytotoxic effects by inducing different types of cellular stress such as genotoxic, proteotoxic, oxidative, and metabolic stress. To overcome these toxic effects, cancer cells activate a network of pathways that enable them to adapt to these cellular stresses. The involvement of HDACs in the response to cellular stress opens the avenue for combining HDACIs with anti-cancer drugs as an effective strategy to overcome the resistance of cancer cells to cancer therapeutics.

Promising findings came from in vitro and in vivo investigations for the combination of HDACIs with anticancer drugs and/or radiotherapy (Table 3). This combination regimen leads to synergistic or additive antitumor effects through different molecular mechanisms [152]. Efficient DNA repair is one of the mechanisms of the resistance of cancer cells to therapy-induced genotoxic stress. Thus, targeting DNA repair proteins represents an effective approach to enhance the response of cancer cells to therapy-induced genotoxic stress. HDACIs were found to selectively deactivate key regulators of DNA repair proteins such as ATM, MRE11, and RAD50 in cancer cells [36]. Thus, Phase I/II clinical trials are currently conducted to investigate the combination of vorinostat with Olaparib (PARP inhibitor) in terms of safety and effectiveness in patients with refractory lymphoma (NCT03259503) and metastatic breast cancer (NCT03742245) [36]. Moreover, HDACIs including TSA, SAHA, MS-275, and OSU-HDAC42 have been reported to sensitize prostate cancer to DNA-damaging agents such as bleomycin, doxorubicin, and etoposide through modulating the acetylation of Ku70 [153]. However, these data warrant further investigations by in vivo study and clinical trials.

Combination therapy involving topoisomerase II inhibitors and HDACIs has been shown to achieve a higher efficacy in the inhibition of topoisomerase II and in the induction of DNA damage [154]. Consequently, a phase I clinical trial combining valproic acid and epirubicin was performed in solid tumors and showed an anti-tumor activity in anthracyclines-resistant patients [155,156]. Moreover, Chen et al. reported that the inhibition of SIRT1 sensitizes lung cancer cells to MK-1775, a selective inhibitor of G2/M checkpoint protein WEE1, resulting in the induction of DNA replication stress and DNA damage [157]. On the other side, this combination strategy can also serve to overcome resistance to HDACIs. The most common resistance mechanism is the increased expression of adenosine-triphosphate-binding cassette (ABC) transporters, which causes drug efflux and induces multidrug resistance, in addition to the elevated levels of cell cycle protein p21 and thioredoxin induced by HDACIs, which lower ROS-mediated DNA damage and contribute to HDACIs resistance [158].

Despite the approved efficacy of HDACIs in hematological malignancies, a limited therapeutic potency is demonstrated against solid tumors when used as single agents [154]. Combining HDACIs as chemosensitizers to other cancer therapeutics showed a great potential in preclinical and clinical trials and may thus represent an avenue to achieve their full therapeutic benefits [159]. As our knowledge has built up to understand how tumors utilize HDACs to overcome cellular stress, rational combination strategies can be implemented in cancer cells. In summary, HDACIs show enormous potential for the treatment of hematological and solid tumors, especially when used in combination with other anticancer agents [136].

**Table 3 ijms-23-08141-t003:** Clinical trials of HDAC inhibitors in combination with other anticancer agents.

HDACIs	Combination Drugs	Clinical Trial Phase	Clinical Trial ID	Cancer Type	Trial Description	Status	References
Vorinostat (SAHA)	Paclitaxel, Trastuzumab, Doxorubicin, Cyclo-phosphamide	I and II	NCT00574587	Breast cancer, gastric cancer	To determine the optimal dose of vorinostat when combined with standard chemotherapy alone (or with trastuzumab when treating HER2-positive cancer).	Completed	[160]
	Doxorubicin hydrochloride	I	NCT00331955	Unspecified adult solid tumor	Vorinostat may help doxorubicin work better by making tumor cells more sensitive to the drug.	Completed	[161]
	Capecitabine and cisplatin	I and II	NCT01045538	Non-small cell lung	Phase 1—maximum tolerated dose, Phase 2—response rate.	Completed	[162]
	Bortezomib	II	NCT00798720	Cancer	The combination showed a weak anti-tumor activity as third-line therapy in NSCLC.	Completed	[163]
Panobinostat (LBH589)	Trastuzumab/ Paclitaxel	I	NCT00788931	HER-2-positive breast cancer, metastatic breast cancer	Combination is well tolerated.	Completed	[164]
	Capecitabine Lapatinib	I	NCT00632489	Lung vancer, head and neck cancer	Combination of Panobinostat and capecitabine is well tolerated at the recommended doses.	Completed	[165]
	Erlotinib	I	NCT00738751	Lung adenocarcinoma	Panobinostat increases the sensitivity of lung adenocarcinoma cells to the antiproliferative effects of erlotinib. (synergism)	Completed	[166]
Belinostat (PXD101)	Doxorubicin	I/II	NCT00878800	Solid tumors and soft tissue sarcomas	No evidence of synergy between belinostat and doxorubicin in terms of objective tumour shrinkage.	Completed	[167]
Valproic Acid	Epirubicin 5-Fluorouracil Cyclo-phosphamide	I	NCT00246103	Advanced neoplasms	Maximum tolerated dose and recommended phase II dose was VPA 140 mg/kg/d for 48 h followed by epirubicin 100 mg/m^2^.	Completed	[156]

In addition to chemotherapy, the use of immune checkpoint inhibitors (ICIs) in cancer has recently remodeled the treatment approaches for solid tumors and hematological malignancies [93]. Currently, many ICIs have been approved by the FDA, such as ipilimumab (anti-CTLA-4); cemiplimab, nivolumab, and pembrolizumab (anti-PD-1); and atezolizumab, durvalumab, and avelumab (anti-PDL) [168,169]. In spite of the promising effects of ICIs, cancer cells still evolve the ability to evade host immune system. Thus, combining ICIs with epigenetic modulators such as HDACIs, which have an immunomodulatory effect, provides a novel strategy that has been shown to produce a beneficial consequence both in vitro and in vivo [170,171]. These benefits include an increase in the immune response and modulation of drug resistance in the tumor microenvironment. A recent study demonstrated that the underlying reason of cancer resistance to ICIs might be due to the presence of immune suppressor cells such as myeloid-derived suppressor cells (MDSCs) in the tumor microenvironment [172]. Interestingly, these immune suppressor cells were found to be targeted by a selective class I HDACI entinostat, which results in augmenting the anti-tumor effect of anti-PD-1 in a mouse model of lung and renal cell carcinoma and subsequently overcoming the resistance to ICIs [173]. In line with this, entinostat was reported to reduce and regulate the MDSC CD40 expression in breast cancer patients [174]. Moreover, the resistance of lymphoma cells to anti-PD-1 drugs was reversed in vitro by administration of a novel class I, IIb, and IV HDACI (OKI-179) [175]. Furthermore, a combination of anti-CTLA-4 and anti-PD-1 antibodies with the pan-HDACI belinostat potentiated the efficacy of these antibodies via upgrading the immune function and reducing tumor volume in a murine hepatocellular carcinoma model [176]. Similarly, a synergistic effect of the triple combination of vorinostat (HDACI) with anti-CTLA-4 and anti-PD-1 was observed in a triple-negative 4T1 breast cancer mouse model through boosting the anti-tumor activity [177]. In melanoma cells, the combination of HDACI with a PD-1 blockade was found to augment tumor immunogenicity [110]. Currently, several clinical trials for the use of ICIs in combination with HDACIs are still going on as listed in Table 4. Collectively, these results rationalize combination therapy between HDACIs and ICIs.

## 6. Conclusions and Future Direction

Since dysregulation of HDACs and manipulated cellular stress phenotypes are major hallmarks of cancer cells; therefore, revealing a connection between HDACs and the response to cellular stress may boost our understanding of the underlying mechanisms of the survival and growth of cancer cells. This might open a new avenue for novel strategies to attack this aggressive disease. However, the existence of a crosstalk between different cellular stress responses has further increased the complexity of understanding the bridge between HDACs and cellular adaption pathways. This review article highlights the advanced molecular mechanisms of the cellular stress response modulated by the differential activity of HDACs isoforms. In addition, we summarize the implications of HDAC activity in cancer growth and in evasion of the cell death pathways and host immunity. Recent clinical trials have exposed the conceptual role of HDACIs and the improved clinical outcomes after their implementation in cancer treatments. Future recommendations necessitate improved understanding of the stress responses adopted by cancer cells for establishing more efficacious therapeutic strategies that might be designed by combining stress-induced therapies with pan HDACIs or isoform-selective HDACIs.

## Figures and Tables

**Figure 1 ijms-23-08141-f001:**
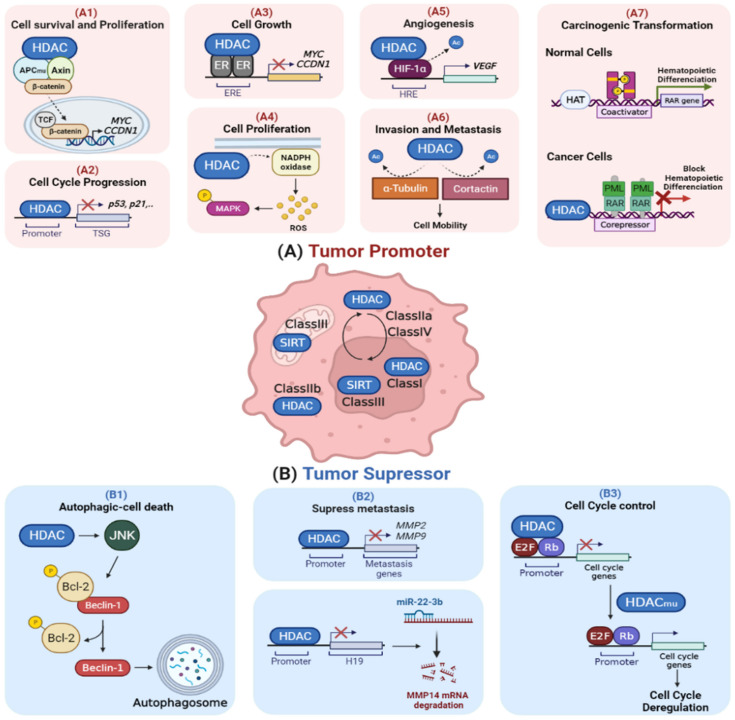
**HDACs as tumor promoters and suppressors.** (**A**) HDACs are overexpressed in cancer, which promotes cellular proliferation and suppresses apoptosis and cell cycle arrest. (A1) Some HDACs (e.g., HDAC2) stabilize the beta-catenin complex, boosting cell survival and proliferation. (A2) HDACs are involved in the disruption of cell cycle checkpoints by obstructing the expression of tumor suppressor genes such as p53 and p21. (A3) Certain HDACs, such as HDAC1, reduce the transcription activity of estrogen receptor-α (ER-α), resulting in growth promotion. (A4) HDACs (e.g., HDAC6) increase the activation of MAPK through inducing the production of reactive oxygen species (ROS) via NADPH oxidase, thus promoting cell proliferation and survival. (A5) Under hypoxia, HDACs (e.g., HDAC1) stabilize HIF-1α through deacetylation, which, in turn, activates the transcription of genes involved in oxygen delivery, energy metabolism, angiogenesis, and apoptosis. (A6) The deacetylation of cell motility proteins (tubulin and cortactin) by HDACs (e.g., HDAC6) drives the progression of a primary tumor to invasion and metastasis. (A7) In hematological malignancies, PML-RAR oncofusion proteins act as altered transcription factors, which aberrantly recruit HDACs to the promoter site of retinoic acid (RA) genes, thus suppressing myeloid differentiation and causing malignant transformation. (**B**) Some HDACs have tumor suppressor activities and they are genetically downregulated in cancer. (B1) HDACs suppress tumor growth through activating JNK-mediated Beclin1 dissociation from Bcl-2 to induce caspase-independent autophagy death. (B2) HDACs (e.g., HDAC10) inhibit invasion and metastasis through reducing the histone acetylation level at the promoter sites of the matrix metalloproteinases (MMP2 and MMP9) genes, thereby suppressing their expression. Additionally, HDACs (e.g., HDAC2) suppress cancer metastasis through inhibiting expression of LncRNA H19, a miR-22-3P sponge that upregulates the expression of MMP14, by histone H3K27 deacetylation at its promoter site. (B3) Some HDACs (e.g., HDAC1) are involved in cell cycle regulation by forming a complex with E2F and RB that represses cell cycle progression genes. Mutations in HDAC1 reduce its recruitment and binding to E2F-regulated promoters, thereby reducing their interaction with retinoblastoma protein (Rb). This results in preventing the repression of cell cycle genes by retinoblastoma (Rb). Abbreviations: APC, adenomatous polyposis coli; TSG, tumor suppressor gene; ER, estrogen receptor; ERE, estrogen response element; ROS, reactive oxygen species; MAPK, mitogen-activated protein kinase; HIF-1α, hypoxia-inducible factor-1; HRE, hypoxia-response element; VEGF, vascular endothelial growth factor; RAR, retinoic acid receptor; PML, promyelocytic leukemia; HAT, histone acetyltransferase; JNK, c-Jun N-terminal kinases; Bcl-2, B-cell lymphoma 2; MMP, matrix metalloproteinases; miR, microRNA; Rb, retinoblastoma.

**Figure 2 ijms-23-08141-f002:**
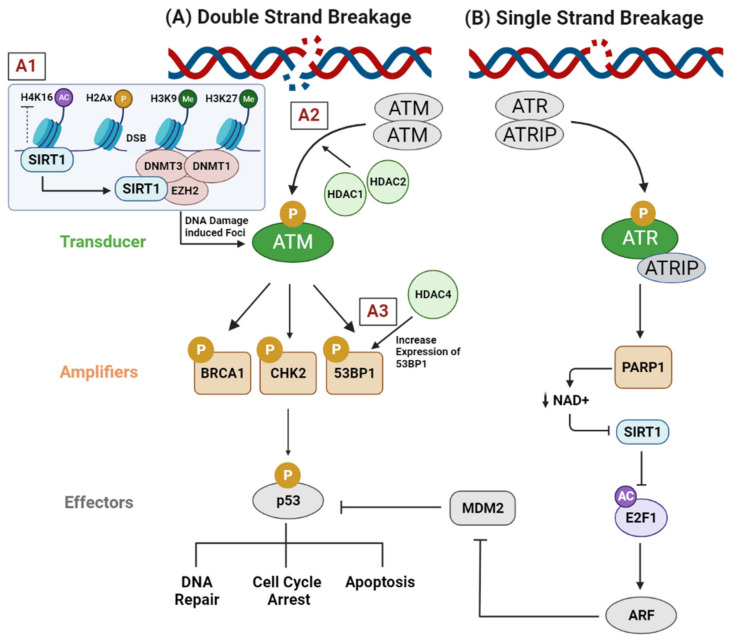
**Role of HDACs in the modulation of DNA repair machinery.** (**A**) After DNA double strand breakage, (A1) SIRT1 is directed towards the DNA damage site to induce chromatin reorganization by recruiting epigenetic machinery including DNMT1, DNMT3B, and EZH2. These epigenetic modifiers induce histone repressive modifications, such as hypoacetylation of H4K16 and methylation of H3K9 and H3K27, that help to establish the compaction of chromatin around the damaged site. (A2) HDAC1 and HDAC2 contribute to ATM activation in several tumor types, thereby enhancing the subsequent phosphorylation of BRCA1, CHK2, and p53, which decreases the susceptibility of DNA breakage. (A3) HDAC4 increases the expression of the tumor suppressor gene 53BP1, a protein involved in the early stages of DNA damage signaling. This signaling cascade facilitates the phosphorylation and activation of p53 protein, which promotes cell cycle arrest to allow DNA repair and/or apoptosis. (**B**) Upon induction of single stand breaks, poly(ADP-ribose) synthesis is catalyzed by PARP1 at unrepaired single break sites, which reduces the activity of NAD+-dependent deacetylase SIRT1 through decreasing the cellular concentration of NAD+. Consequently, the acetylation of E2F1 is maintained, which activates the transcription of ARF, which inhibits MDM2, a negative regulator of p53. This allows p53 to exert its tumor suppressor transcriptional regulation or/and to induce apoptosis. Abbreviations: DNMT, DNA methyltransferase 1; DNMT3B, DNA methyltransferase 3B; EZH2, enhancer of zeste homolog 2; ATM, ataxia-telangiectasia mutated; BRCA1, breast cancer gene 1; CHK2, checkpoint kinase 2; 53BP1, p53-binding protein 1; ATR, ataxia-telangiectasia and Rad3-related; ATRIP, ATR interacting protein; PARP1, poly(ADP-ribose) polymerase 1; NAD+, nicotinamide adenine dinucleotide; E2F1, E2F transcription factor 1; ARF, alternative reading frame; MDM2, mouse double minute 2.

**Figure 3 ijms-23-08141-f003:**
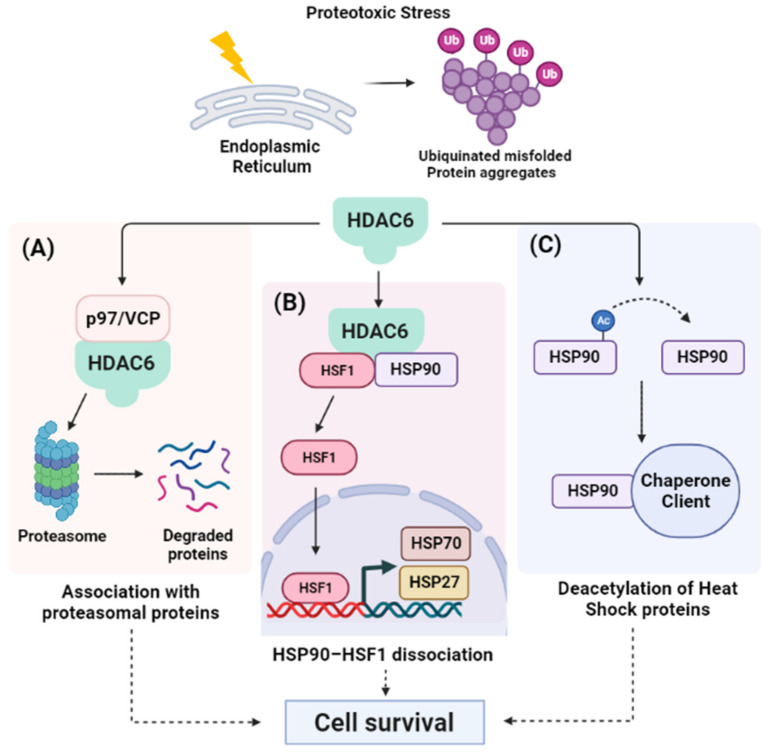
**HDAC6 regulates the response to misfolded protein aggregates.** Under proteotoxic stress, cancer cells evolve an intricate set of signaling to allow the cell to respond to the presence of misfolded proteins within the endoplasmic reticulum (ER). HDAC6 is considered a master regulator in misfolded protein processing through three different pathways: (**A**) HDAC6 forms a complex through its ZnF-UBP domain with proteasomal proteins such as p97/VCP that binds to polyubiquitinated proteins, facilitating their proteasomal degradation; (**B**) In unstressed cells, HDAC6 usually exists in complex with an inactive HSF1 and HSP90. Upon ER stress, this complex is dissociated and the liberated HSF1 activates the expression of heat shock genes (HSP70 and HSP27). These heat shock proteins are essential components of the cell machinery that are required for the proper folding of proteins and the degradation of damaged proteins to protect against the adverse effects of proteotoxic stress; and (**C**) HDAC6 deacetylates HSP90, enhancing its chaperone activity, and facilitates the recruitment of other chaperones to reduce the level of unfolded proteins. At the end, upgrading the chaperone capacity in processing misfolded proteins by HDAC6 promotes cell survival. Abbreviations: Ub, ubiquitin; VCP, valosin-containing protein; HSF1, heat shock transcription factor 1; HSP90, heat shock protein 90; HSP70, heat shock protein 70; HSP27, heat shock protein 27.

**Figure 4 ijms-23-08141-f004:**
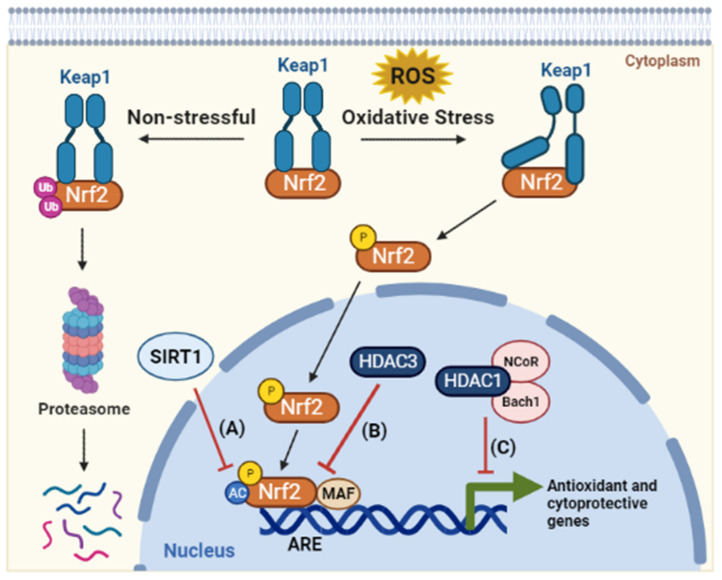
**HDACs coordinate the cellular response to oxidative stress.** HDACs coordinate the cellular response to oxidative stress. As a defense mechanism against oxidative stressors, Nrf2 acts as a redox-sensitive transcription factor. Under homeostatic conditions, Keap1 sequesters Nrf2 in the cytoplasm, resulting in its degradation by proteasome. When the cells are subjected to oxidative stress, Nrf2 is activated through its dissociation from the Keap1 complex. Then, the Nrf2 protein is transported to the nucleus, where it is acetylated and dimerized with the MAF protein at the antioxidant reactive element (ARE) site in the promoters of antioxidant genes. Through this process, antioxidant genes are activated to scavenge excess reactive oxygen species (ROS) and maintain mitochondrial function. In response to oxidative stress, HDACs act as a negative regulator of the Nrf2 pathway. (**A**) SIRT1 inhibits Nrf2 acetylation, which reduces its binding to ARE and thereby obstructs Nrf2-mediated gene expression. (**B**) HDAC3 is recruited to the ARE site and induces a histone hypoacetylation state, hence repressing ARE-dependent gene expression. (**C**) HDAC1 forms a complex with Bach1, which competes with Nrf2 for the binding to transcription cofactor MAF in oxidative-stress-response genes. Bach1 acts as a functional inhibitor of Nrf2 by forming a complex with nuclear co-repressor NCoR. Abbreviations: Nrf2, NF-E2–related factor 2; Keap1, Kelch-like ECH associated protein 1; ROS, reactive oxygen species; MAF, musculoaponeurotic fibrosarcoma; ARE, antioxidant responsive element; Bach1, BTB domain and CNC homolog 1; NCoR, nuclear receptor corepressor 1.

**Figure 5 ijms-23-08141-f005:**
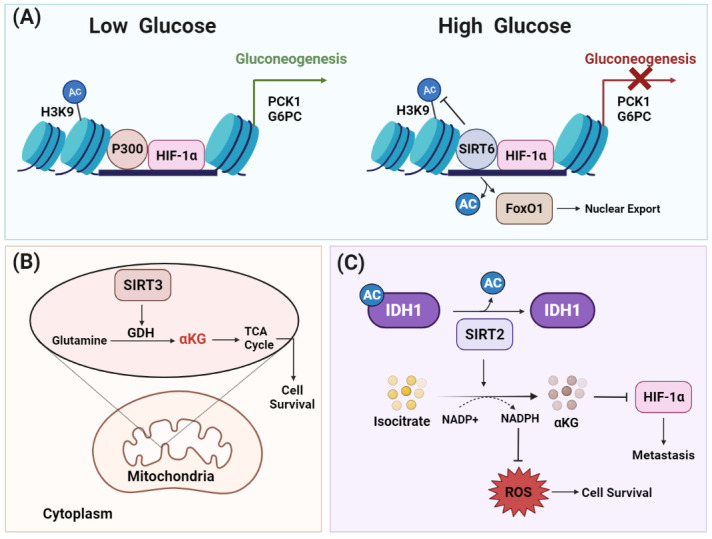
**HDACs elicit metabolic reprogramming to support cell survival under metabolic stress.** Different HDACs are involved in reglueing several metabolic pathways. (**A**) Under low glucose levels, histone acetyltransferase p300 acts as coactivator for HIF-1α to induce transcription of genes involved in gluconeogenesis. Under high glucose levels, SIRT6 inhibits gluconeogenesis by acting as a co-repressor of HIF-1α transcriptional activity by deacetylating H3K9 at the promoter sites of its target glycolytic genes such as PCK1 and G6PC, which are a rate-limiting enzymes in glucose synthesis. SIRT6 deacetylates the transcription factor FoxO1, which results in its export to the cytoplasm. Excluding FoxO1 from the nucleus reduces the expression of PCK1 and G6PCm thereby inhibiting gluconeogenesis. (**B**) The mitochondrial lysine deacetylase SIRT3 promotes glutamine flux to the TCA cycle via glutamate dehydrogenase, which, in turn, shields the cancer cells from metabolic stress and maintains cell survival. (**C**) SIRT2 deacetylates IDH1 at lysine 224 and stimulates its metabolic activity. IDH catalyzes the decarboxylation of isocitrate to produce alpha-ketoglutarate (αKG), which is important for the hydroxylation and degradation of HIF-1α. Consequently, SIRT2-dependent IDH1 deacetylation inhibits metastasis as well as decreasing reactive oxygen species (ROS) levels by producing NADPH, an ultimate donor for ROS-detoxifying enzymes. Abbreviations: HIF-1α, hypoxia-inducible factor-1; FoxO1, Forkhead box protein O1; PCK1, phosphoenolpyruvate carboxykinase 1; G6PC, glucose-6-phosphatase catalytic subunit; IDH1, isocitrate dehydrogenase 1; αKG, alpha-ketoglutarate; ROS, reactive oxygen species; GDH, glutamate dehydrogenase; TCA, tricarboxylic acid.

**Table 1 ijms-23-08141-t001:** Classification of Histone deacetylases (HDACs) according to the down-target substrates and their cellular localization.

Class	Member	Location	Substrates	Inhibitors	References
**Class I**	HDAC 1 HDAC 2 HDAC 3 HDAC 8	Nuclear	Histones	Belinostat Vorinostat Panobinostat Entinostat Valproic acid Romidepsin	[14]
**Class IIa**	HDAC4 HDAC5 HDAC7 HDAC9	Nuclear/cytoplasmic	Histones	Belinostat Vorinostat Panobinostat Entinostat Valproic acid	[15]
**Class IIb**	HDAC 6 HDAC 10	Nuclear/cytoplasmic	Histones; α-tubulin; Hsp90	Belinostat Vorinostat Panobinostat	[16]
**Class III**	Sirtuins Sir2	Nuclear/cytoplasmic	Mitochondrial Histones; Tubulin; p53; TAF	Nicotinamides	[17]
**Class IV**	HDAC 11	Nuclear	Rpd3 protein	Belinostat Vorinostat Panobinostat Entinostat Romidepsin	[18]

**Table 2 ijms-23-08141-t002:** HDAC inhibitors in clinical studies.

HDACIs	Clinical Trial Phase	Clinical Trial ID	Cancer Types	Trial Description	Status	References
MS-275 (Entinostat)	I	NCT00020579	Refractory solid tumors and lymphoid	Well tolerated at a dose of 6 mg/m^2^, administered weekly with food for 4 weeks every 6 weeks	Completed	[142]
Romidepsin (Depsipeptid)	I	NCT00053963	Solid tumors	Well tolerated in children with refractory solid tumors	Completed	[143]
	II	NCT00106613	Renal cell carcinoma	Did not have sufficient activity.	Completed	[144]
	I	NCT00077337	Colorectal cancer	Romidepsin at dose of 13 mg/m^2^ as a 4 h iv infusion on days 1, 8, and 15 of a 28-day cycle was ineffective in treatment of metastatic colon cancer	Completed	[133]
Panobinostat (LBH589)	II	NCT00667862	Hormone refractory prostate cancer	Panobinostat did not show a sufficient level of clinical activity to undergo further investigation in CRPC	Completed	[145]
	II	NCT00425555	Cutaneous T-cell lymphoma	Panobinostat was generally well tolerated with no major safety concerns.	Completed	[146]
	I	NCT00412997	Solid tumors	Doses well tolerated up to 20 mg in Japanese patients	Completed	[147]
Belinostat (PXD 101)	I	NCT01273155	Adult primary hepatocellular carcinoma and advanced adult primary liver cancer	Increased belinostat exposure accompanied hepatic dysfunction	Completed	[148]
	I and II	NCT00321594	Localized unresectable adult primary liver cancer and recurrent adult primary liver cancer	Phase I—belinostat tolerated at maximum dose of 1200 mg/m^2^/day Phase II—will start with 1200 mg/m^2^/day	Completed	[149]
Vorinostat	I	NCT00097929	Relapsed diffuse large B-cell lymphoma	Limited activity against relapsed DLBCL	Completed	[150]
	II	NCT00132067	Primary peritoneal cavity recurrent ovarian epithelial cancer	Vorinostat well tolerated with minimal activity as a single agent	Completed	[135]

**Table 4 ijms-23-08141-t004:** Clinical trials of HDAC inhibitors in combination with immune checkpoint inhibitors.

HDACIs	Immune Checkpoint Inhibitor	Clinical Trial Phase	Clinical Trial ID	Cancer Types	Trial Description	Status	References
Vorinostat (SAHA)	Pembrolizumab	I/II	NCT02638090	Lung cancer/stage IV NSCLC	-	Recruiting	[178]
		I/Ib	NCT02619253	Renal cell carcinoma and urinary bladder neoplasms	-	Recruiting	[179]
Entinostat	Atezolizumab	I/II	NCT02708680	Breast cancer	Combination therapy resulted in more toxicity	Completed	[180]
	Nivolumab	I/II	NCT03838042	Central nervous system tumors, solid tumors	-	Recruiting	[181]

## Data Availability

Not applicable.

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
