# Peer review of "The Role of HDACs in the Response of Cancer Cells to Cellular Stress and the Potential for Therapeutic Intervention"

_ijms, 2022, doi:10.3390/ijms23158141_

Round 1
Reviewer 1 Report
This review covers a great deal of territory regarding an important cluster of bioactive agents.
In general the text flows well and is clear but there is certainly room for improvement as the following comments reveal.
1. The wording of the bottom line of the Abstract is indistinct. Similarly at the bottom of Figure 1.
2. There are inappropriate choice of words on occasions such as "wretched' (line 69), "outrageous" (157), inconsistent (158, 167), sighted (169), add 'up" to give "to sum up" (436), "the" to be inserted before tumor (484), "of" to be removed (501, 709), spelling of mitochondria is spelled incorrectly (Figure 5), above mentioned (581), "grant" should be replaced by "allow" (651) and lines 691 / line 692 needs rephrasing.
Apart from these considerations, this review is recommended for publication.
Author Response
We thank the reviewer for his/her valuable comments. All suggested modifications/revisions have been introduced into the revised version of the manuscript
Reviewer 2 Report
The manuscript is well structured and contains a good overview of the field of HDAC inhibitors and their molecular mechanisms especially regarding cellular stress. In some places the authors could add more recent references and important results. Especially in paragraph 4 where immunotherapies are described. Here an entry on the interaction of the PDL1 system with different HDAC subtypes would be appropriate.
HDAC5 modulates PD-L1 expression and cancer immunity via p65 deacetylation in pancreatic cancer. Zhou Y, Jin X, Yu H, Qin G, Pan P, Zhao J, Chen T, Liang X, Sun Y, Wang B, Ren D, Zhu S, Wu H. Theranostics. 2022 Jan 31;12(5):2080-2094.
HDAC inhibition potentiates anti-tumor activity of macrophages and enhances anti-PD-L1-mediated tumor suppression. Li X, Su X, Liu R, Pan Y, Fang J, Cao L, Feng C, Shang Q, Chen Y, Shao C, Shi Y. Oncogene. 2021 Mar;40(10):1836-1850.
A selective HDAC8 inhibitor potentiates antitumor immunity and efficacy of immune checkpoint blockade in hepatocellular carcinoma. Yang W, Feng Y, Zhou J, Cheung OK, Cao J, Wang J, Tang W, Tu Y, Xu L, Wu F, Tan Z, Sun H, Tian Y, Wong J, Lai PB, Chan SL, Chan AW, Tan PB, Chen Z, Sung JJ, Yip KY, To KF, Cheng AS. Sci Transl Med. 2021 Apr 7;13(588):eaaz6804.
Author Response
Thank you very much for the valuable revision
References have been modified as suggested and the three recommended references have been added to the article